# Quality Determination of the Saudi Retail Banking System and the Challenges of Vision 2030

**Mohammad Ishfaq \*, Heitham Al Hajieh and Majed Alharthi** 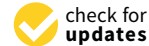

Finance Department, College of Business, King Abdulaziz University, P.O. Box 344, Rabigh 21911, Saudi Arabia; Haawadh@kau.edu.sa (H.A.H.); mdalharthi@kau.edu.sa (M.A.)

**\*** Correspondence: mishfaq@kau.edu.sa

**Abstract:** Vision 2030 of the Kingdom of Saudi Arabia (KSA) requires numerous national and multinational organizations to re-engineer themselves to achieve the required targets for the upturn of the Saudi economy. In this respect, the quality of indigenous goods and services has been the biggest challenge to satisfy consumers of Saudi businesses. The banking and finance sector, specifically, has a great deal of responsibility to put in place a strong financial system that is capable of attracting capital from both local and foreign investors. SERVQUAL, with the five conventional dimensions—tangibility, reliability, responsiveness, assurance and empathy—offers a great deal of flexibility in modifying the model to the specific requirements of a service in carrying out gap analysis. In this context, we have applied SERVQUAL by adding two new dimensions—functional and technical—to the conventional five dimensions. We applied SERVQUAL using a "performance-only approach" to identify quality gaps present in the services of national and multinational banks. Our analysis shows that gaps exist in the service quality—both in national and multinational banking systems. We therefore present weighted gap scores to assist service managers in setting up priorities to improve the quality of their services. This study suggests that there is much to be done to improve retail banking quality and gain customers' confidence, both from within and outside the KSA.

**Keywords:** SERVQUAL; Saudi Arabian retail banking system; KSA Vision 2030

## 1. Introduction

Saudi Arabian economy is shifting from oil export to the development of an indigenous industry that delivers goods and services of a high quality. The underlying objective is to become less reliant on oil exports and produce more indigenous products and services with exportable potential. The government considers that the high-quality innate services in areas like travel, insurance, healthcare, education, retailing and banking would help in overcoming the trade deficit by reducing import costs and less reliance on an imported workforce.

Announced in 2016, the Saudi Vision 2030 (Kingdom of Saudi Arabia 2019) shows that the Kingdom of Saudi Arabia (KSA) realizes that the quality of goods and services always remains a prime concern for customers. This paradigm is not new. Lassar et al. (2000) and Zahorik and Rust (1992) have previously argued that service quality is an essential element in establishing a satisfying relationship with valued customers. Zahorik and Rust (1992) also link service quality with profitability, and we believe that service quality creates an advantage that leads to high performance and financial benefits. Supporting other similar studies, Zhang et al. (2019) demonstrate that service quality results in consumer satisfaction and consumer loyalty that renders financial benefits; conversely, poor quality has been found to cause financial distress (Palmer 1995; Zeithaml et al. 1996). Within this context, our study concentrates on a highly complex and sensitive Saudi Arabian retail banking system and its related quality issues.

The Saudi Arabian financial system is well integrated under the Saudi Arabian Monetary Authority (SAMA, with the role of the central bank). SAMA, established in 1957, regulates retail banks, private investment programs, specialized lending institutions and the stock market (Saudi Arabia Country Commercial Guide 2018). Alongside commercial banking, SAMA monitors financial service infrastructure in the Kingdom through the Capital Market Authority (CMA). CMA has licensed 91 foreign and local companies. These companies are responsible for providing investment and brokerage services (Meyer-Reumann 1995).

SAMA's main objective is to reduce transaction costs and enhance transaction security while achieving consumer satisfaction. SAMA shares its developments with all licensed banks in the KSA to ensure they adhere to best performance practices and deliver ideal service (Meyer-Reumann 1995). Bearing in mind the sensitive nature of the financial sector's responsibilities and its impact on the economy in general, and consumer satisfaction in particular, this study has been designed to apply the SERVQUAL model, modified by adding technical and functional dimensions to the conventional five dimensions of quality. The objectives of (a) applying an integrated SERVQUAL model after adding technical and functional dimensions to the conventional five dimensions of the quality, (b) making use of integrated SERVQUAL to determine whether there exist any quality gaps in retail banking services and, if so, what is their impact on customer satisfaction and (c) comparing quality dimensions between national and multinational banks operating in the Kingdom to explore whether any dichotomy exists in the service quality of the two sectors. While applying SERVQUAL, we agree with the theorem that a relationship exists between service quality and financial gains or losses, as presented by Lassar et al. (2000).

## 2. Literature Review

Quality is vital to customer satisfaction (Spreng and Mackoy 1996). Literature on the notion of quality refers back to the 1960s when Vroom (1964) presented expectancy theory. Further research, such as Oliver (1980), Lovelock (1981), Lehtinen and Lehtinen (1982) and Gronroos (1982), led to the development of SERVQUAL by Parasuraman et al. (1988), a scale of measuring quality gaps. SERVQUAL has been applied in numerous research studies to measure gaps in service quality with various modifications (Carman 1990; Babakus and Boller 1992; Cronin and Taylor 1992). Accordingly, our prime objective has been to modify SERVQUAL by adding two dimensions to the existing five quality dimensions for measuring quality gaps in the Kingdom's banking sector.

The quality of goods and services is generally determined by a difference between consumer perception and the expectation of a service. If expectation exceeds the perception, the quality is considered to have gaps, causing consumer dissatisfaction (Lewis and Mitchell 1990). Slack et al. (2010) view quality as a consistent conformance to customer expectations, as presented in Figure 1.

Quality gaps are measured to have a clear understanding of a specific quality dimension to plan and prioritize improvements in the service quality. This is essential to gain an advantage over competitors (Jensen and Markland 1996; Lassar et al. 2000). With such an ideology, Ishfaq et al. (2016) applied SERVQUAL to determine quality gaps in healthcare insurance in Saudi Arabia. The analysis, which used the five dimensions of SERVQUAL, showed a negative gap with respect to "reliability", indicating that practitioners should target that specific area of the service quality. The airline industry is another example where SERVQUAL has been applied to dig out quality gaps and bring about improvements (Aydin and Yildirim 2012; Pakdil and Aydın 2007; Rezaei et al. 2018). In the banking sector, recent studies used SERVQUAL to rank competitors and estimate gap elasticity with the objective of hammering out gaps in service quality (Dinçer et al. 2019; Kumar et al. 2018).

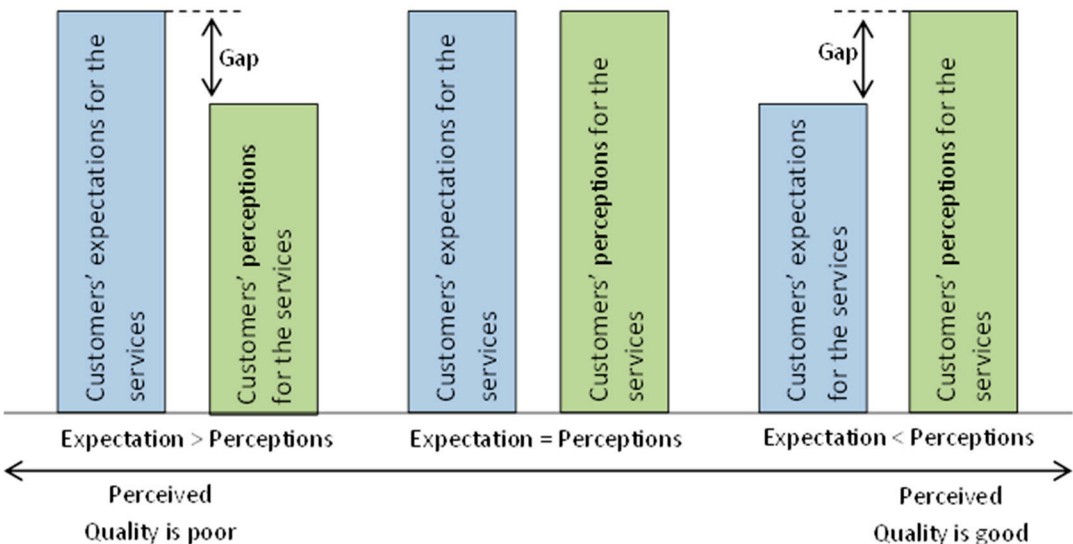

**Figure 1.** Quality perception and expectation.

*SERVQUAL Model*

Parasuraman et al. (1985) presented comprehensive research to measure service quality by defining "service quality as the gap between customers' expectations of service and experience". A standard SERVQUAL model originating from this study and has been widely used in measuring service quality. The most prominent studies are of measuring quality in hotel industries (Gržinić 2007), in public services (Brysland and Curry 2001) and in psychometric and diagnostic criteria (Parasuraman et al. 1994). In all studies, the common approach has been to identify discrepancies between consumers' perceptions of quality and their expectations, generally around five dimensions, i.e., reliability, responsiveness, assurance, empathy and tangibles.

The scope of this study is essentially to determine gaps in the quality of banking services using modified SERVQUAL. Improved quality has many advantages. For example, the best quality assures consumer satisfaction, as presented by Grönroos (1983). Similarly, Rabbani et al.'s study in 2016 examined and found that there is a strong relationship between quality, consumer satisfaction and profitability in banking operations. Further evidence is available from the literature that service quality and consumer satisfaction are highly correlated (Oliver 1993; Spreng and Mackoy 1996; Zhang et al. 2019). Zeithaml et al. (1996) argue that service quality has significant implications for financial gains or losses. Our argument is also in line with the findings of Zhang et al. (2019) and Zeithaml et al. (1996), who claim that identifying quality gaps and closing them enhances consumers' satisfaction and profitability, as also claimed by Grönroos (1983), Lassar et al. (2000) and Rabbani et al. (2016).

## 3. Problem Statement

Vision 2030 creates opportunities for national and multinational businesses to invest in Saudi Arabia. It is therefore imperative for the Saudi banking system to be seen as a trusted custodian of potential investors' capital. In the past, the banking sector came under global criticism for financial crises causing enormous losses to millions. A lack of service reliability has been the major concern that has triggered consumer dissatisfaction (Gilbert and Wong 2003; Spreng and Mackoy 1996; Oliver 1993).

The increasing complexity and multidimensionality of banking services is making consumers more sensitive to service delivery and they expect that banks will provide a fool-proof system of quality assurance. By "multidimensionality", we mean the numerous financial products on offer alongside conventional banking services, such as checking accounts, current and fixed deposits, automated tellers, credit cards, short- and long-term financing, utility bill payments and governmental transactions. These financial services are offered both by national and multinational banks under the umbrella of the Saudi banking system. In addition to the conventional banking system, all banks in the KSA, either

national or multinational, are encouraged to follow the Islamic ethos, thus making the banking service spectrum quite deep and wide. With such a broad mix of financial services and products, both local and foreign banks are competing for a fair share in the Saudi financial market.

How objectively the share of the financial market is achieved is a question that could be best answered by measuring service quality and linking it to consumer satisfaction plus financial gains or losses. We presume that consumer satisfaction is highly correlated with service quality, as Lassar et al. (2000) argue. Anouze et al. (2019) have also argued that consumer satisfaction has positive impact on banks' performances in terms of market share. With the resolve of Anouze et al. (2019) and Lassar et al. (2000) in mind, we argue that the Saudi banking system must ensure that there are no gaps in its services. Accordingly, we plan to measure the quality of the banking sector by applying the SERVQUAL model, together with technical and functional measures of the quality (Lassar et al. 2000).

*3.1. Study Objectives*

This study presents gap analysis to determine the quality of the banking services in Saudi Arabia. Primary data were collected from 300 respondents. The study is organized in a way to achieve the following objectives:

(1)　Provide justification for applying an integrated SERVQUAL model, in the light of discussion presented elsewhere, to measure banking service quality in a broad spectrum.
(2)　Apply the modified SERVQUAL model to identify quality issues in the services of Saudi Arabian banks.
(3)　Examine and compare quality gaps between national and multinational banks operating in the KSA.
(4)　Suggest remedies to resolve quality issues and provide dimensions for further research.

*3.2. Hypotheses*

According to the objectives we have designed, we developed three hypotheses with the following resolves: (1) statistical justification for modifying SERVQUAL, (2) determine quality gaps using extended SERVQUAL and (3) compare service quality between national and multinational banking systems operating in KSA. The hypotheses are:

**Hypothesis 1 (H1).** *There are additional factors/dimensions other than the five conventional dimensions of the SERVQUAL model that influence the quality of banking services operating in the KSA.*

**Hypothesis 2 (H2).** *Quality gaps exist in almost all quality dimensions and their respective items—resulting customer dissatisfaction.*

**Hypothesis 3 (H3).** *There are quality gaps in all or some quality dimensions between national and multinational banks in the KSA.*

## 4. Research Methodology

This research is organized over five stages, as explained in the following sub-sections.

*4.1. What Is the Best Approach for Measuring Quality Gaps?*

The literature suggests that there are five approaches for measuring gaps in service quality (Gilbert and Wong (2003). These are: (1) expectancy disconfirmation, (2) performance only, (3) technical and functional dichotomy, (4) service quality versus service satisfaction and (5) attribute importance. Researchers can adopt any approach according to their research objectives. We have selected a

performance-only approach for our study because it was convenient to collect data and carryout gap analysis.

Prior to Gilbert and Wong's (2003) study, Teas (1993) developed an evaluated performance (EP) model to overcome difficulties with the gap models presented by Grönroos (1983) and Parasuraman et al. (1988). A detailed discussion on the difficulties examined by Teas (1993) with the gap model is beyond the scope of this study. However, their study indicated that the perception minus expectation (P minus E) model has questionable validity, specifically the "measurement validity of the expectation (E)" and, consequently, an alternative performance-only model was developed to address the problems of the traditional framework (Teas 1993).

In this respect, we decided to apply the evaluated performance (EP) model, considering that consumers always expect an ideal level of service, hence there was no question on the measurement validity of the expectation (E). Accordingly, our questionnaire asked respondents to score their experience of the bank performance on a Likert scale of 1 to 5. The data collected on performance were used to determine gaps between consumers' experiences of the service quality against the ideal expectations. This is in line with Slack et al.'s (2010) model, presented in Figure 1, in which a gap score of consumer perception (CP) minus consumer expectation (CE) = 0 means no gap in the service quality and, conversely, a score of −4 means absolute failure in the delivery of the quality services. Having clarified the use of the EP approach in this study, the following section explains the reasons for adding technical and functional quality constructs to the five conventional quality dimensions of the SERVQUAL model.

*4.2. Technical and Functional Quality Measures*

Gilbert and Wong (2003) argue that both technical and functional constructs lead to consumer satisfaction. In 1983, Grönroos presented an alternative model of these measurements, describing functional quality as how a service, for example, procedures in a hospital, and technical quality are what is being provided. Gilbert and Wong (2003) specify that technical quality is based on product characteristics, whereas functional quality is concerned with the relationship between a service provider and a consumer.

The literature suggests that functional and technical constructs have not been used as much as the traditional SERVQUAL dimensions (Lassar et al. 2000); however, some researchers have found technical and functional constructs quite relevant to a specific service. For example, Baker and Lamb (1994) argue that in the case of architectural design, "customers rely on the functional-based dimension of service quality because they may not have the knowledge or skills to evaluate [the] technical-based dimension". Similarly, in the case of accountancy services, Higgins and Ferguson (1991) report that "clients evaluated both functional and technical dimensions of the quality; the functional dimension, however, seemed to carry more weight". Richard and Allaway (1993) reported that in the case of pizza delivery "both technical and functional dimensions explained more of the variation in customer choice behaviour than the functional dimension alone".

Here, we argue that in retail banking, the functional and technical dimensions are equally important because the banking sector heavily relies on off-site services through the internet and automated teller machines. Nevertheless, the role of branch-based services cannot be ignored, in which a blend of technology and procedures influence service quality.

We also further explored whether additional dimensions could influence quality, leading to consumer satisfaction. Edward S. Mason (1930), cited in Lassar et al. (2000), presented the structure–process–performance model. Lassar et al. (2000) used this model as a theoretical backdrop and incorporated technical and functional quality dimensions to determine customer loyalty and satisfaction. The application of the structure–process–performance model is beyond the scope of this research. Therefore, our main focus is a gap analysis using SERVQUAL to evaluate the quality of retail banking services. In this study, we added technical and functional constructs to the five traditional constructs of SERVQUAL to determine quality gaps concerning two banking systems operating in

the KSA: (1) overall quality gaps, comprising both national and multinational banks, using the data collected through a questionnaire (Appendix A) from 300 respondents and (2) quality gaps of Saudi national versus multinational banks operating in the KSA.

### 4.3. Questionnaire Development

A questionnaire was developed to collect primary data from customers to test our hypotheses (Appendix A). We ensured that the questionnaire was clear so respondents could easily understand it and provide correct answers. To facilitate quality responses, we also added an Arabic translation to the questionnaire.

To test the content validity of the questionnaire, it was submitted to eight experts and academics in the marketing and finance fields to ensure that the content and structure were understandable and workable and to verify whether potential respondents would be able to complete the questionnaire. Moreover, the initial version of the questionnaire was reviewed by MBA students who are experts in this field to ensure content validity.

### 4.4. Data Collection

Data were collected randomly from 300 users of retail banking services in Saudi Arabia. The questionnaire was distributed to about 400 users with a target of 350 respondents. The data were collected through trained research assistants under the supervision of a senior researcher to ensure that the data collected were from reliable sources and the maximum responses were received. We ultimately received responses from 300 customers, which made up 86% of the expected responses. We considered this response quite satisfactory.

The participants rated the questionnaire items on a Likert scale of 1 to 5, where 5 stands for the highest rating and 1 indicates the lowest rating for the quality of service received (Gržinić 2007).

### 4.5. Data Tabulation

The collected data were input into SPSS software for both analysis and interpretation. The statistical tests were then carried out to assess the reliability and quality of the data. Finally, a factor analysis was carried out and gaps were determined.

### 5. Statistical Models and Data Quality Interpretation

The data collected were analyzed and interpreted in light of the hypotheses and the objectives of the study. The sample size was determined using a formula from Creative Research Systems (1982), which indicated that 300 responses is an acceptable sample size for the analysis.

SPSS is a major statistical tool for establishing the reliability of collected data. First, we carried out a factor analysis to determine the importance of technical and functional quality dimensions in SERVQUAL. Second, we carried out a gap analysis based on the seven dimensions. A factor analysis was performed by applying the following model (Pakdil and Aydın 2007):

$$Ft_{pi} = \sum_{j=1}^{k} w_j \times x_{jpi} \tag{1}$$

where $Ft_{pi}$ is the perception score for each factor, $w_j$ is the factor loading of the $j$th item, $j = 1, 2, \ldots, k$, $k$ is the number of items included in the $t$th factor and $x_{jpi}$ is the $i$th respondent's perception score for the $j$th item.

The SERVQUAL model was developed by Parasuraman et al. (1985) using the following equation:

$$SERVQUAL = CP - CE, \tag{2}$$

where CP is consumers' perception and CE is consumers' expectation

The average gap score, $G_i$, for each dimension was calculated using the following model:

$$G_i = \frac{1}{N} \sum_{i=1}^{N} (CP - CE) \tag{3}$$

where $N$ is the number of items in each dimension, $CP$ is the consumer perception and $CE$ is the consumer expectation (ideal score in our case).

Cronin and Taylor (1992) developed the SERVPERF model that implicitly assesses customers' experiences based on the same attributes as the SERVQUAL model. Hence, we deduce that measuring service quality just by evaluating a consumer's overall feeling towards the service would make a significant contribution. Our approach is further supported by Teas (1993), offering an evaluated performance model (EP) that "measures the gap between perceived performance and the ideal amount of a feature not customers expectation".

We may briefly discuss here the difference between SERVQUAL and SERVPREF. The SERVQUAL model holds that when service perception exceeds the expected service, it implies that service quality is less satisfactory. Alternatively, where perception is less than the expected service, this means that service quality is more than satisfactory (Parasuraman et al. 1988). Their model is based on a set of 22 elements around five constructs of quality dimensions. Since their model is operationalized by identifying gaps between customers' expectations and perceptions of performance, the quality measurement scale is comprised of a total of 44 items (22 for expectations and 22 for perceptions).

Cronin and Taylor (1992) questioned the conceptual basis of the SERVQUAL scale and found it confusing with regards to service satisfaction. They argued that the expectation (E) component of SERVQUAL should be discarded and instead the performance (P) component alone be used. They proposed what is referred to as the "SERVPERF" scale (Jain and Gupta 2004).

### 5.1. Profile Analysis of Banking Service Users

The Saudi population heavily relies on banking services. According to one estimate provided by the International Monetary Fund in 2016, there are 992.59 bank accounts per 1000 adults in the Saudi Arabian population (TheGlobalEconomy.com 2016). This implies that our survey of 300 respondents covers about 31% of Saudi Arabian bank account holders per 1000 people. The population profile, from whom the data were collected, is given in Table 1.

**Table 1.** Demographic characteristics of the sample.

| Demographic Variables | Demographic Characteristics | Frequencies in % | Cumulative % |
|---|---|---|---|
| Age | Up to 25 | 34.5 | 34.5 |
| | 26–35 | 34.9 | 69.4 |
| | 36–50 | 25.3 | 94.7 |
| | 50+ | 5.3 | 100 |
| Gender | M | 73.4 | 73.4 |
| | F | 26.6 | 100 |
| Marital Status | Single | 41.8 | 41.8 |
| | Married | 49 | 90.8 |
| | Divorced | 6.2 | 97 |
| | Widow | 3 | 100 |
| Education | High School | 17.4 | 17.4 |
| | Graduate | 52 | 69.4 |
| | Master's | 22 | 91.4 |
| | PhD | 8.6 | 100 |

**Table 1.** *Cont.*

| Demographic Variables | Demographic Characteristics | Frequencies in % | Cumulative % |
|---|---|---|---|
| Profession | Student | 32.6 | 32.6 |
| | Un-employed | 3.9 | 36.5 |
| | FT Employed | 47.7 | 84.2 |
| | Executives | 11.5 | 95.7 |
| | Self-employed/business | 4.3 | 100 |
| Income | <5000 | 36.8 | 36.8 |
| | 5000–9900 | 21.4 | 58.2 |
| | 10,000–14,900 | 31.2 | 89.5 |
| | 15,000–19,900 | 2.3 | 91.8 |
| | 20,000+ | 8.2 | 100 |
| Bank | Saudi National | 68.3 | 68.3 |
| | Non-Saudi/Multinational | 31.7 | 100 |

The surveyed population is a good mix of age and income groups, educational levels, gender, employment and usage of both Saudi national and multinational banks. Table 1 also shows that the consumer population is relatively young and economically vibrant, with a high percentage of graduates. Of the surveyed population, 68.3% use Saudi banks. The reliability of data was tested using Cronbach's alpha ($\alpha$) measure, as follows.

## 5.2. Cronbach's $\alpha$: Data Reliability Test

To establish the reliability of the collected primary data, we carried out a Cronbach's $\alpha$ test and present the results in Table 2.

**Table 2.** Cronbach's $\alpha$: Data reliability test.

| Reliability Statistics | |
|---|---|
| Cronbach's Alpha | Number of Items |
| 0.972 | 30 |

The reliability test shown in Table 2 proves that the data are quite reliable with a Cronbach's $\alpha$ score of 0.972. This suggests that there is no error in 97.2% of the sample but there could be an error in the remaining 2.8%. According to Parasuraman et al. (1988), Cronbach's $\alpha$ ranges from 0 to 1, meaning no reliability (0) to perfect internal reliability (1) of the primary data. Accordingly, the results in Table 2 show a high level of data reliability for each of the seven dimensions. Bartlett's test further confirms a high level of correlation among the data variables with a chi-square value of 4.269 at the 0.000 significance level. This confirms that a factor analysis is quite relevant for this sample. Because our aim is to study gaps in the service provision, we further focused on the seven dimensions of service quality and analyzed the Cronbach's $\alpha$ for each dimension and impact on $\alpha$ if an item is deleted, as shown in Table 3.

Table 3 shows that if any item is deleted from the analysis, the Cronbach's $\alpha$ decreases, which means that the item is highly significant and should not be eliminated from the analysis. However, when items P5 and P23 were deleted, the Cronbach's $\alpha$ slightly increased, suggesting that if these items are removed from the analysis, it would cause a slight increase in the reliability. Nevertheless, we consider this increase quite insignificant and therefore decided not to delete P5 and P23 from our analysis.

**Table 3.** Banking service quality constructs and their reliability tests.

| Dimensions | Cronbach's Alpha for Dimension | Cronbach's Alpha If Item Is Deleted | Item |
|---|---|---|---|
| Reliability | 0.890 | 0.870 | P1 |
| | | 0.853 | P2 |
| | | 0.853 | P3 |
| | | 0.854 | P4 |
| | | 0.893 | P5 |
| Responsiveness | 0.863 | 0.828 | P6 |
| | | 0.832 | P7 |
| | | 0.817 | P8 |
| | | 0.826 | P9 |
| Assurance | 0.851 | 0.811 | P10 |
| | | 0.829 | P11 |
| | | 0.810 | P12 |
| | | 0.792 | P13 |
| Empathy | 0.839 | 0.803 | P14 |
| | | 0.837 | P15 |
| | | 0.786 | P16 |
| | | 0.754 | P17 |
| Tangibles | 0.851 | 0.784 | P18 |
| | | 0.742 | P19 |
| | | 0.850 | P20 |
| Technical | 0.813 | 0.770 | P21 |
| | | 0.770 | P22 |
| | | 0.814 | P23 |
| | | 0.773 | P24 |
| | | 0.754 | P25 |
| Functional | 0.889 | 0.868 | P26 |
| | | 0.864 | P27 |
| | | 0.868 | P28 |
| | | 0.867 | P29 |
| | | 0.858 | P30 |

*5.3. Applying a Factor Analysis for Justifying the Seven Quality Measures*

A factor analysis is normally used in SERVQUAL applications because data for SERVQUAL are based on the Likert scale, which cannot be interpreted as averages or standard deviations to express statistical results. A factor analysis, however, provides weighted scores by factor loadings and is based on the computation of intercorrelations among variables. Inspecting the correlation matrix, we found that most of the variables are positively correlated, but it is difficult to derive a complete and clear understanding of their relationship (Ishfaq 1993). Therefore, the variables were reduced to a smaller set of derived variables, called factors (Siddique et al. 2013).

*5.4. Factor Loading, Eigenvalues and Internal Consistency*

According to Hair et al. (1998), an equal number of factors can be computed as the number of variables; nevertheless, only those factors whose eigenvalues are 1 or higher should be extracted. In our case, we extracted seven factors whose eigenvalues were higher than 2 and the variance percentage was more than 50%. Cronbach's $\alpha$ for each dimension was over 0.80. Therefore, we conclude that adding two additional dimensions—technical and functional—to the SERVQUAL model made our analysis statistically robust and meaningful in the determination of gaps in banking services. Pakdil and Aydın (2007) used eight dimensions, and eight factors were extracted in the case of the airline quality gap determination. In the financial sector of Saudi Arabia, we have not seen any evidence of

applying the SERVQUAL model with additional service quality dimensions. We, therefore, added two more dimensions to the SERVQUAL model and carried out a factor analysis, as presented in Table 4.

**Table 4.** Factor loading analysis (principal component).

| | Component | | | | | | |
|---|---|---|---|---|---|---|---|
| | **Factor 1** | **Factor 2** | **Factor 3** | **Factor 4** | **Factor 5** | **Factor 6** | **Factor 7** |
| ServiceReliabilityE1 | 0.782 | | | | | | |
| ServiceReliabilityE2 | 0.568 | | | | | | |
| ServiceReliabilityE3 | 0.818 | | | | | | |
| ServiceReliabilityE4 | 0.619 | | | | | | |
| ServiceReliabilityE5 | 0.84 | | | | | | |
| ResponsivenessE6 | | 0.653 | | | | | |
| ResponsivenessE7 | | 0.672 | | | | | |
| ResponsivenessE8 | | 0.819 | | | | | |
| ResponsivenessE9 | | 0.608 | | | | | |
| AssuranceE10 | | | 0.608 | | | | |
| AssuranceE11 | | | 0.519 | | | | |
| AssuranceE12 | | | 0.826 | | | | |
| AssuranceE13 | | | 0.625 | | | | |
| EmpathyE14 | | | | 0.837 | | | |
| EmpathyE15 | | | | 0.915 | | | |
| EmpathyE16 | | | | 0.629 | | | |
| EmpathyE17 | | | | 0.533 | | | |
| TangiblesE18 | | | | | 0.562 | | |
| TangiblesE19 | | | | | 0.599 | | |
| TangiblesE20 | | | | | 0.668 | | |
| TechnicalE21 | | | | | | 0.782 | |
| TechnicalE22 | | | | | | 0.837 | |
| TechnicalE23 | | | | | | 0.915 | |
| TechnicalE24 | | | | | | 0.84 | |
| TechnicalE25 | | | | | | 0.818 | |
| FunctionalE26 | | | | | | | 0.608 |
| FunctionalE27 | | | | | | | 0.625 |
| FunctionalE28 | | | | | | | 0.826 |
| FunctionalE29 | | | | | | | 0.819 |
| FunctionalE30 | | | | | | | 0.608 |
| % of Variance | 13.591 | 12.953 | 12.112 | 11.9 | 10.918 | 10.307 | 9.594 |
| Cumulative % | 13.591 | 26.543 | 38.655 | 50.555 | 61.473 | 71.78 | 81.374 |

Table 4 shows that with the additional quality dimensions—technical and functional—the percentage of cumulative variance has become significantly high, suggesting that SERVQUAL should be modified to broaden the scope of gap analysis. Therefore, we accept H1 and modify the SERVQUAL model for this study.

## 6. Gap Analysis for the Entire Sample and Discussion of Results

We believe consumer satisfaction relies on service quality, as has been discussed and proven by Lassar et al. (2000) in their study. Having accepted their argument, we determine the gaps in Saudi banking service quality by applying the SERVQUAL model for all seven dimensions. Based on Equation (2), the gap score for each item, and based on Equation (3), unweighted average score for each dimension, was calculated and is presented in Table 5.

**Table 5.** Gap calculations.

| S.N | Statement | CP | CE | CP − CE |
|-----|-----------|-----|-----|---------|
| | **Service Reliability** | | | |
| P1 | Bank provides service as promised | 3.953333 | 5 | −1.04667 |
| P2 | Always active in solving problem | 3.883333 | 5 | −1.11667 |
| P3 | Service provided correctly at the first time | 3.856667 | 5 | −1.14333 |
| P4 | Service provided at the time promised | 3.873333 | 5 | −1.12667 |
| P5 | Bank keeps accurate and updated records | 4.163333 | 5 | −0.83667 |
| | Total | 19.73 | 25 | −5.27 |
| | Mean Service Reliability Score | 3.946 | 5 | −1.054 |
| | **Responsiveness** | | | |
| P6 | Customers informed when service will be given | 3.77 | 5 | −1.23 |
| P7 | Gives service promptly | 3.766667 | 5 | −1.23333 |
| P8 | Bank employees are willing to help customers | 3.833333 | 5 | −1.16667 |
| P9 | Responds well to customer requests | 3.883333 | 5 | −1.11667 |
| | Total | 15.25333 | 20 | −4.74667 |
| | Mean Responsiveness Score | 3.050667 | 4 | −1.18667 |
| | **Assurance** | | | |
| P10 | Bank employees are trustworthy | 3.943333 | 5 | −1.05667 |
| P11 | Transactions carried out Safely | 4.23 | 5 | −0.77 |
| P12 | Employees are always polite | 3.983333 | 5 | −1.01667 |
| P13 | Employees found with having knowledge answering questions | 3.926667 | 5 | −1.07333 |
| | Total | 16.08333 | 20 | −3.91667 |
| | Mean Assurance Score | 4.020833 | 5 | −0.97917 |
| | **Empathy** | | | |
| P14 | Customers are given individual attention | 3.793333 | 5 | −1.20667 |
| P15 | Bank working hours are convenient | 3.833333 | 5 | −1.16667 |
| P16 | Bank has customers' best interests at heart | 3.803333 | 5 | −1.19667 |
| P17 | Bank understands specific need of customers | 3.803333 | 5 | −1.19667 |
| | Total | 15.23333 | 20 | −4.76667 |
| | Mean Empathy Score | 3.808333 | 5 | −1.19167 |
| | **Tangibles** | | | |
| P18 | Bank had modern-looking and up-to-date equipment | 3.97 | 5 | −1.03 |
| P19 | Best appearance of physical facilities | 3.96 | 5 | −1.04 |
| P20 | Best appearance of employees | 3.79 | 5 | −1.21 |
| | Total | 11.72 | 15 | −3.28 |
| | Mean Tangibles Score | 3.906667 | 5 | −1.09333 |
| | **Technical** | | | |
| P21 | Service is provided with ease | 3.916667 | 5 | −1.08333 |
| P22 | Attention to individual needs | 3.81 | 5 | −1.19 |
| P23 | Ease in contacting bank services providers | 3.793333 | 5 | −1.20667 |
| P24 | Accurate record keeping giving results | 4.053333 | 5 | −0.94667 |
| P25 | Providing information correctly | 3.776667 | 5 | −1.22333 |
| | Total | 19.35 | 25 | −5.65 |
| | Mean Technical Score | 3.87 | 5 | −1.13 |
| | **Functional** | | | |
| P26 | Employees are trustworthy and keep confidentiality | 3.943333 | 5 | −1.05667 |
| P27 | Employees available to answer questions | 3.926667 | 5 | −1.07333 |
| P28 | Employees are always courteous and friendly | 3.983333 | 5 | −1.01667 |
| P29 | Competence in explaining services and policies | 3.833333 | 5 | −1.16667 |
| P30 | Understand requests and give good responses | 3.883333 | 5 | −1.11667 |
| | Total | 19.57 | 25 | −5.43 |
| | Mean Functional Score | 3.914 | 5 | −1.086 |

Table 5 shows that none of the SERVQUAL items has a score equal to zero. This means that there are quality gaps in each item of the service, causing consumer dissatisfaction. Nevertheless, we can say certain items do have a trivial gap between CP and CE, suggesting that with a little attention these gaps could be removed. The items with trivial gaps are P5 (bank keeps accurate and updated records, under service reliability dimension) and P11 (transaction carried out safely, under assurance dimension), and P24 (accurate record keeping giving results, under technical dimension).

Unweighted average scores in Table 6 reflect that assurance is the only dimension which has a trivial gap, suggesting a comparatively low level of dissatisfaction (−0.9792).

**Table 6.** Unweighted average scores.

| Dimension | Score |
|---|---|
| Service Reliability | −1.054 |
| Responsiveness | −1.187 |
| Assurance | −0.979 |
| Empathy | −1.192 |
| Tangibles | −1.093 |
| Technical | −1.130 |
| Functional | −1.086 |
| Mean | −1.103 |

This analysis therefore guides managers to focus upon areas of trivial quality gaps and set the standard up with minimum efforts. A further way of examining the performance of each dimension is to determine the weights which banking service consumers would give to the seven dimensions in SERVQUAL. We could not collect such data through our questionnaire, in order to avoid complications in the questionnaire. In the absence of that, a full-blown Delphi approach could have helped in determining the weights of each dimension through a consensus among experts. Delphi is primarily a forecasting technique for "harnessing and organizing judgements in complex problem requiring intuitive interpretation of evidence or informed guesswork" (Thangaratinam and Redman 2005). We consider that a Delphi-like technique could be utilized in this research to determine weights by asking experts what weight they would give to the seven dimensions. We presumed that our faculty members, having substantial knowledge and experience of the banking service in the Kingdom, would appropriately resolve the weighting issue. Accordingly, we asked faculty members to give each dimension of SERVQUAL a score out of 100. This analysis helped to present gap scores in an understandable scale, as well as assist service managers to set priorities in attempting to improve service quality. Delphi normally requires three rounds, however, we did it in a single round due to a lack of time. Importance weights and mean weighted scores are presented in Table 7.

**Table 7.** Weighted average scores.

| Dimension | Unweighted Score | Importance Weightage | Weighted Score |
|---|---|---|---|
| Service Reliability | −1.054 | 20.62 | −22% |
| Responsiveness | −1.187 | 13.59 | −16% |
| Assurance | −0.979 | 17.94 | −18% |
| Empathy | −1.192 | 5.67 | −7% |
| Tangibles | −1.093 | 4.93 | −5% |
| Technical | −1.130 | 19.67 | −22% |
| Functional | −1.086 | 17.59 | −19% |
| Mean | −1.103 | 100.00 | −16% |

The weighted average score given to each dimension in Table 7 suggests that consumers are sensitive with reference to service reliability, functional and technical dimensions that could cause a high level of dissatisfaction. From this analysis, we infer that the gap analysis not only identifies the

level of consumers' satisfaction or dissatisfaction in each item of the SERVQUAL, but it also provides information on which dimension of the SERVQUAL is more important to customers, so improvements could be carried out accordingly. Hence, we accept hypothesis $H_2$, that there are significant gaps in service quality causing dissatisfaction among customers. This calls for bank management to resolve quality issues.

## 7. Gap Comparison

From Table 1, we find that 68.3% of the respondents use Saudi banks and 31.7% use non-Saudi multinational banks. Accordingly, we carried out a gap analysis separately for Saudi and non-Saudi banks to examine whether multinational banks provide better services compared to the Saudi national banks. The comparative gap statistics of Saudi national, non-Saudi multinational and combined Saudi and non-Saudi banks are presented in Table 8.

**Table 8.** Comparative gap statistics between Saudi national, non-Saudi multinational and combined banking systems.

| Non-Saudi | G1 | G2 | G3 | G4 | G5 | G6 | G7 | AV. |
|---|---|---|---|---|---|---|---|---|
| Mean | −1.109 | −1.153 | −0.984 | −1.153 | −1.133 | −1.12 | −1.065 | −1.103 |
| Standard Deviation | 0.891 | 0.949 | 0.893 | 0.967 | 1.016 | 0.854 | 0.877 | 0.921 |
| Variance | 0.793 | 0.9 | 0.797 | 0.936 | 1.033 | 0.73 | 0.768 | 0.851 |
| Skewness | −1.601 | −1.098 | −1.602 | −1.111 | −1.228 | −1.554 | −1.399 | −1.37 |
| Kurtosis | 3.127 | 1.085 | 3.061 | 1.137 | 1.15 | 3.028 | 2.607 | 2.171 |
| One-Sample *t*-test | 0 | 0 | 0 | 0 | 0 | 0 | 0 | 0 |
| **Saudi** | | | | | | | | |
| Mean | −1.028 | −1.202 | −0.977 | −1.207 | −1.0748 | −1.135 | −1.096 | −1.1028 |
| Standard Deviation | 0.889 | 1.011 | 0.893 | 0.975 | 0.731 | 0.856 | 0.933 | 0.898 |
| Variance | 0.791 | 1.023 | 0.798 | 0.95 | 0.534 | 0.732 | 0.87 | 0.814 |
| Skewness | −1.024 | −0.795 | −1.294 | −0.847 | −1.095 | −1.041 | −1.045 | −1.02 |
| Kurtosis | 0.613 | −0.096 | 1.574 | 0.169 | 1.277 | 0.945 | 0.75 | 0.747 |
| One-Sample *t*-test | −0.005 | −0.006 | 0.003 | −0.005 | 0 | 0.006 | 0.006 | 0 |
| **Saudi and Non-Saudi** | | | | | | | | |
| Mean | −1.054 | −1.187 | −0.979 | −1.19 | −1.093 | −1.13 | −1.086 | −1.103 |
| Standard Deviation | 0.891 | 0.992 | 0.893 | 0.973 | 1.007 | 0.855 | 0.915 | 0.895 |
| Variance | 0.793 | 0.985 | 0.798 | 0.946 | 1.014 | 0.731 | 0.838 | 0.804 |
| Skewness | −1.204 | −0.883 | −1.392 | −0.929 | −1.066 | −1.203 | −1.147 | −1.119 |
| Kurtosis | 1.394 | 0.193 | 1.983 | 0.424 | 0.571 | 1.534 | 1.205 | 1.06 |
| One-Sample *t*-test | 0 | 0.006 | −0.003 | 0 | −0.006 | 0 | 0 | 0.006 |

Table 8 shows one-sample *t*-tests of all three types of banking systems and it examines whether the mean gaps are statistically different in each dimension of each bank type. The results show, that for each dimension of each bank type, the mean gap is not statistically different. Furthermore, we see that the standard deviations, variances, skewness (data negatively tailed) and kurtosis (data relatively scattered around the mean, resulting in a flat shape) of the three bank types are almost similar. Therefore, we conclude that the Saudi banking sector, in totality, lacks service quality, no matter how trivial it may be. Therefore, we accept $H_3$, that there are quality gaps in all quality dimensions of national and multinational banks operating in the KSA.

## 8. Discussion

Strong consumer confidence in banking services is essential for the successful functioning of the banking sector. Consumer confidence, however, necessitates an ideal service having no quality gaps. In this study, we offer SERVQUAL with modifications to measure quality gaps, enabling managers to identify areas requiring attention. Although, initially, SERVQUAL was applied in the marketing sector,

in recent times, SERVQUAL has been applied in numerous industries, such as traveling, healthcare, airlines, information technology (Nimako et al. 2012), hotels and insurance. SERVQUAL can be applied in any industry because it offers a great deal of flexibility.

SERVQUAL has a great deal of flexibility, as many researchers have applied this model successfully with modifications, according to the specific requirements of their study. For example, Pakdil and Aydın (2007) modified SERVQUAL by adding new quality dimensions—technical and functional—to broaden the scope of their study in investigating the impact of service quality on consumer satisfaction and business performance. Zhang et al. (2019) use eight dimensions to measure omnichannel retail quality. Following Zhang et al. (2019) and Pakdil and Aydın (2007), we modified SERVQUAL and carried out factor analysis to examine its suitability in the case of KSA's banking services. Accordingly, we applied this model using seven dimensions of quality. Statistical analysis has provided a strong basis for applying modified SERVQUAL to measure gaps in the quality of banking services. This study further concentrates on comparing quality gaps between Saudi national, non-Saudi multinational and combined Saudi and non-Saudi banks. This is also a unique approach, suggesting the use of SERVQUAL in comparing quality services between two sectors. We have found that there are identical gaps in the service quality of both the sectors that need to be addressed to win consumers' confidence and satisfaction. This study also provides a guideline for the regulatory authorities of KSA to ensure banking services work to win consumers' confidence and help in attracting capital—as Vision 2030 requires. More studies could be carried out by studying the impact of demographic characteristics on service quality, making use of a Geographical Information System (GIS) to understand the quality gaps between various localities.

There are certain limitations that were not addressed due to the time and resource constraints, for example, the size of sample, the impact of demographic characteristic on service quality and differences in quality gaps among localities. We believe this study will motivate academics to extend studies in these directions.

**Author Contributions:** Conceptualization, M.I.; Data curation, H.A.H.; Formal analysis, H.A.H.; Funding acquisition, M.I.; Investigation, M.I. and M.A.; Methodology, M.I.; Project administration, M.A.; Software, H.A.H.; Validation, M.A.; Writing—original draft, M.I.; Writing—review and editing, M.I., H.A.H. and M.A. All authors have read and agreed to the published version of the manuscript.

**Acknowledgments:** This work was funded by the Deanship of Scientific Research, King Abdulaziz University, Jeddah, under Grant No. 309-849-1439. We are grateful to the Deanship of Scientific Research, King Abdulaziz University, Jeddah for their funding this project.

**Conflicts of Interest:** The authors declare no conflict of interest.

## Appendix A

**Questionnaire for Banking Quality**

استبيان جودة المصرفية

**Personal Data**

المعلومات الشخصية

| Age: | Up to 25 years | | 26–35 years | | 36–50 years | | 50+ | |
|------|------|------|------|------|------|------|------|------|
| العمر | تحت 25 سنة | | 26–35سنة | | 36–50سنة | | 50سنة فأكثر | |

| Gender: | Male | | Female | |
|------|------|------|------|------|
| الجنس | ذكر | | أنثى | |

| Nationality: | Saudi | | Non-Saudi | |
|------|------|------|------|------|
| الجنسية | سعودي | | غير سعودي | |

| Marital Status: | Single | | Married | | Widow | | Divorced | |
|---|---|---|---|---|---|---|---|---|
| الحالة الاجتماعية | أعزب | | متزوج | | أرملة | | مطلق | |

| Education: | Under Grade 10 | | Grade 12 | Graduation | | Master | | PhD | |
|---|---|---|---|---|---|---|---|---|---|
| التعليم | أولى ثانوي فأقل | | ثالثة ثانوي فأقل | بكالوريوس | | ماجستير | | دكتوراة | |

| Profession: | Student | | Self Employed/Business | | Employed Fulltime Non-Executive | |
|---|---|---|---|---|---|---|
| العمل | طالب | | عمل حر | | موظف غير تنفيذي | |
| | Employed Fulltime Executive | | Unemployed | | Housewife | |
| | موظف تنفيذي | | غير موظف | | ربة منزل | |

| Your Monthly Income: | >SAR5000 | | 5000–9900 | | 10,000–14,900 | | 15,000–19,900 | | 20,000+ | |
|---|---|---|---|---|---|---|---|---|---|---|
| الدخل الشهري | أقل من 5000 ريال | | 5000–9900 | | 10,000–14,900 | | 15,000–19,900 | | 20,000+ | |

| Your Main Bank: | Saudi National Bank | | Multi-National in KSA | | Saudi/Foreign JV | | Saudi Islamic Bank | |
|---|---|---|---|---|---|---|---|---|
| مصرفك الرئيسي | مصرف سعودي محلي | | مصرف متعدد الجنسيات | | مصرفي سعودي/مصرف أجنبي | | مصرف سعودي اسلامي | |

SERVQUAL Questionnaire Service Perceived

ادراك استبيان نموذج خدمة الجودة

| S.N التسلسل | Statement البيان Service Reliability/ثقة الخدمة | Strongly Disagree لا أوافق بشدة 1 | Disagree لا أوافق 2 | Neutral محايد 3 | Agree موافق 4 | Strongly agree أوافق بشدة 5 |
|---|---|---|---|---|---|---|
| P1 | Bank provides service as promised يقدم المصرف الخدمة كما وعد | | | | | |
| P2 | Always active in solving problem المصرف نشيط في حل المشكلة | | | | | |
| P3 | Service provided correctly at the first time يقدم المصرف الخدمة بشكل صحيح في أول مرة | | | | | |
| P4 | Service provided at the time promised يقدم المصرف الخدمة في الوقت كما وعد | | | | | |
| P5 | Bank keeps accurate and updated records المصرف يحافظ على السجلات بطريقة دقيقة ومحدثة | | | | | |

| S.N التسلسل | Statement البيان Service Reliability/ثقة الخدمة | Strongly Disagree لا أوافق بشدة 1 | Disagree لا أوافق 2 | Neutral محايد 3 | Agree موافق 4 | Strongly agree أوافق بشدة 5 |
|---|---|---|---|---|---|---|
| Responsiveness/الاستجابة | | | | | | |
| P6 | Customers informed when service will be given المصرف يخبر العملاء متى الخدمة ستكون مقدمة | | | | | |
| P7 | Gives service promptly المصرف يعطي الخدمة فوراً | | | | | |
| P8 | Bank employees willing to help customers موظفي المصرف لديهم الرغبة لمساعدة العملاء | | | | | |
| P9 | Responds well to customer requests المصرف يجيب على استفسارات العملاء | | | | | |
| Assurance/الضمان | | | | | | |
| P10 | Bank employees are trustworthy موظفي المصرف موثوقين | | | | | |
| P11 | Transactions carried out safely يوجد أمان في العمليات المصرفية | | | | | |
| P12 | Employees are always polite موظفي المصرف ذوي خلق عالي | | | | | |
| P13 | Employees found with having knowledge answering questions موظفي المصرف لديهم معرفة للاجابة على الأسئلة | | | | | |
| Empathy/التعاطف | | | | | | |
| P14 | Customers are given Individual attention يتم الاهتمام الشخصي بالعميل | | | | | |
| P15 | Bank working hours are convenient ساعات عمل المصرف مناسبة | | | | | |
| P16 | Bank has customers best interest at heart لدى العملاء ولاء عالي تجاه المصرف | | | | | |
| P17 | Bank understands specific need of customers المصرف يفهم احتياجات العملاء | | | | | |
| Tangibles/الملموسات | | | | | | |
| P18 | Bank had modern looking and up-to-date equipment المصرف يمتلك شكل عصري وأدوات محدثة | | | | | |
| P19 | Best appearance of physical facilities المصرف يمتلك أفضل تسهيلات ملموسة | | | | | |
| P20 | Bank appearance of employees موظفي المصرف يظهروا بأفضل شكل | | | | | |
| **Technical** | | | | | | |
| P21 | Service is provided with ease يتم توفير الخدمة بسهولة | | | | | |
| P22 | Attention to individual need يتم الاهتمام بالحاجات الفردية | | | | | |
| P23 | Ease in contacting bank service providers سهولة الاتصال بمزودي الخدمات المصرفية | | | | | |

| S.N التسلسل | Statement البيان Service Reliability/ثقة الخدمة | Strongly Disagree لا أوافق بشدة 1 | Disagree لا أوافق 2 | Neutral محايد 3 | Agree موافق 4 | Strongly agree أوافق بشدة 5 |
|---|---|---|---|---|---|---|
| Responsiveness/الاستجابة | | | | | | |
| P24 | Accurate record keeping giving results يتماستخدام سجلات دقيقة لحفظ المعلومات والنتائج | | | | | |
| P25 | Providing information correctly يتم تقديم المعلومات بشكل صحيح | | | | | |
| **Functional** | | | | | | |
| P26 | Employees are trustworthy and keep confidentiality الموظفون جديرون بالثقة ويحافظون على السرية | | | | | |
| P27 | Employees available to answer questions الموظفين جاهزون للإجابة على الأسئلة | | | | | |
| P28 | Employees are always courteous and friendly الموظفون دائمًا مهذبون وودودون | | | | | |
| P29 | Competence in explaining services and policies يتم شرح الخدمات والسياسات من قبل متخصصين | | | | | |
| P30 | Understand requests and give good response يتم فهم طلبات العملاء ويتم إعطاء استجابة جيدة | | | | | |

Thank you for your cooperation

شكراً لتعاونك

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
