# Peer review of "Quality Determination of the Saudi Retail Banking System and the Challenges of Vision 2030"

_ijfs, doi:10.3390/ijfs8030040_

Round 1
Reviewer 1 Report
Authors should enhance problem statements and literature review with more relevant studies.
Authors should make more clear the conclusions/results in the hypotheses they present.
Author Response
Point by point responses to reviewer 1 comments:
Point 1: English language and style are fine/minor spell check required
Response 1: Spelling checks have been made.
These are in line 3,
Section 1, para 2 line 2, section 1, para 4 line 3
Section 2, para 3 line 2
Section 5, para 1 line 1
Section 5.2 line 9
Section 6, para 5 line 6
Acknowledgement line 1
Point 2: Does the introduction provide sufficient background and include all relevant references? Can be improved.
Response 2:
Section 1: Deleted lines 1 and 2 and added line 2 and 3. We believe this would be fine with the reviewer 1.
Point 3: Is the research design appropriate? Must be improved
Response 3:
Section 4 line 1
Section 4.1 line 1, 2, 4, 5
Section 4.3 added para 2
Section 4.4, deleted line 1 and 2 and added line 2 and 3.
Point 4: Are the methods adequately described? Yes
Response 4: Thanks no need to make changes
Point 5: Are the results clearly presented? Yes
Response 5: Thanks no need to make changes
Point 6: Are the conclusions supported by the results? Can be improved
Response 6: Please see section 8 which has been changed. Reviewer 2 had also asked for improvements in this section..
Point 7: Authors should enhance problem statements and literature review with more relevant studies
Response 7:
Section 2, some changes have been made and changed figure 1 also as required by Reviewer 2.
Section 3, para one edited
Section 3, para 2 edited, para 3 deleted
Section 3, last para edited and new reference added, lines 10 to 14 deleted
Point 8: Authors should make clear the conclusions/results in the hypotheses they present.
Response 8: Conclusion section has been revised. This was also required by Reviewer 2.
Reviewer 2 Report
This paper integrated both SERVQUAL model with functional and technical dimensions to assess customers satisfaction level with banking sector in KSA. I am privileged to have reviewed this particular manuscript. The subject matter is the core capillary of any healthy business. However, the manuscript as present needs work to convince the audience about its relevance and suitability. Below are some of the comments to help the author better the manuscript.
Major Comments:
- Authors are advised to give enough motivation to include extra two dimensions to SERQUAL model.
- I believe that the research objective has to be positioned better with respect to the current literature in banking service quality. In any case any contribution has to be identified over relevant literature.
- Research objectives should be same through the paper (introduction (page 1, study objectives page 4 and page 5).
- Researches are advised to add the source of the scale items, and it is not clear how they addressed the validity of Arabic version (translation).
- It is not clear why the authors apply their study to Jeddah and Rabigh districts rather than the Capital of KSA.
- Third paragraph of page 6 “Cronin and Taylor (1992) developed the SERVPERF model ‘by dropping expectations”. This statement is not correct as the SERVPREF measures the preferences of customers rather than their expectation.
- The authors are advised to add one paragraph about the SERVQUAL model.
- The authors are advised to give enough motivation to the proposed hypotheses.
- The authors are advised to add one paragraph about the difference between SERVQUAL model, which is a gap measurement approach and SERVPREF model which is a performance-based measurement approach.
- Section 5.2: Cronbach’s α: Data Reliability the results presented in this section is not correct.
- Page 9 first paragraph is not clear.
- Table 4: it is clear that the first five dimension explained % of Variance is 77.4% but when we added the 6th and 7th dimensions the percentage is dropped down to 58.07% and 69.43%. Therefore, authors are advised to give a strong motivation to include these two dimensions.
- It is not clear how the authors used Delphi methods and the number of rounds to get the final weights.
- Table 8 is not clear, what is the differences between this table and third part of table 9.
- Table 9: the discussion of this table is not correct.
- It is not clear how the authors tested the third hypotheses.
- It is not clear how the authors tested the proposed hypotheses.
- Section 8 discussion: Line 3-4 this conclusion is not related to this study?
- Section 8: is not clear.
- Authors are advised to up to date the literature review section.
Minor Comments:
- Title: should be change to fit the paper context, the Quality Assurance concept is not discussed through the paper.
- Keywords: gap analysis should be deleted as SERVQUAL is called gap model; quality assurance should be deleted too.
- Summary: Line 14: the word tackle should be replaced by improve.
- Page 6 last paragraph is not clear.
- Authors are advised to double check the figures of table 5.
- Table 6: the presented figures are the average and not the weights.
Author Response
Responses
Point 1: Moderate English changes required
Response 1: Almost all areas of the paper have been looked into and necessary changes have been made. These can be seen through track changes.
Point 2: Does the introduction provide sufficient background and include all relevant references? Can be improved.
Response 2: Please see note 1a and 2a
Point 3: Is the research design appropriate? Yes
Response 3: No changes needed, thanks
Point 4: Are the methods adequately described? Yes
Response 4: No changes needed, thanks
Point 5: Are the results clearly presented? Can be improved
Response 5: Please see tables 2 to 8 appropriate changes have been made
Point 6: Are the conclusions supported by the results? Can be improved
Response 6: Please see paragraph following Table 4, paragraph 4 of section 6 and last paragraph of section 6. Also see paragraph section 6 following Table 8.
Other Major Comments
Point 1: Authors are advised to give enough motivation to include extra two dimensions to SERQUAL model.
Response 1: Please see section 4.2. We believe reviewer would find it appropriate.
Point 2: I believe that the research objective has to be positioned better with respect to the current literature in banking service quality. In any case any contribution has to be identified over relevant literature.
Response 2: Please see section 3.1. Changes have been made accordingly.
Point 3: Research objectives should be same through the paper introduction (page 1, study objectives page 4 and page 5).
Response 3: Please line 9 and 10 of Abstract. Also see study objectives
Point 4: Researches are advised to add the source of the scale items, and it is not clear how they addressed the validity of Arabic version (translation).
Response 4: Please see para 2 of section 4.3
Point 5: It is not clear why the authors apply their study to Jeddah and Rabigh districts rather than the Capital of KSA.
Response 5: Please see line 1,2 & 3 of section 4.4
Point 6: Third paragraph of page 6 “Cronin and Taylor (1992) developed the SERVPERF model ‘by dropping expectations”. This statement is not correct as the SERVPREF measures the preferences of customers rather than their expectation.
Response 6: Please see section 5, paras 3 & 4 (old para 3 is deleted)
Point 7: The authors are advised to add one paragraph about the SERVQUAL model.
Point 7: Please section 2.1, two paragraphs added to explain SEVERQUAL model.
Point 8: The authors are advised to give enough motivation to the proposed hypotheses
Response 8: Please see section 3.2 replaced by edited version
Point 9: The authors are advised to add one paragraph about the difference between SERVQUAL model, which is a gap measurement approach and SERVPREF model which is a performance-based measurement approach.
Response 9: Please see section 5, paragraphs 4 and 5
Point 10: Section 5.2: Cronbach’s α: Data Reliability the results presented in this section is not correct.
Response 10: Please see Table 2 section 5.2
Point 11: Page 9 first paragraph is not clear.
Response 11: Please see Study Objective para 1 and objective one and also see para one section 3.2
Point 12: Table 4: it is clear that the first five dimension explained % of Variance is 77.4% but when we added the 6th and 7th dimensions the percentage is dropped down to 58.07% and 69.43%. Therefore, authors are advised to give a strong motivation to include these two dimensions.
Response 12: Please see Table 4 with new results and the paragraph following Table 4.
Point 13: It is not clear how the authors used Delphi methods and the number of rounds to get the final weights.
Response 13: Please see paragraph following Table 6
Point 14: Table 8 is not clear, what is the differences between this table and third part of table 9.
Response 14: Table 8 has been deleted and Table 9 is now Table 8. Please see paragraph following Table 8.
Point 15: Section 8 discussion: Line 3-4 this conclusion is not related to this study?
Response 15: Discussion section has been completed revised.
Point 16: Section 8: is not clear
Response 16: Section 8 is completely revised
Point 17: Authors are advised to up to date the literature review section
Response 17: Literature review in section 2 has been moderately revised and section 2.1 has been added.
Minor Comments:
Point 1: Title: should be change to fit the paper context, the Quality Assurance concept is not discussed through the paper.
Response 1: Title has been changed by replacing Assurance with Determination
Point 2: Keywords: gap analysis should be deleted as SERVQUAL is called gap model; quality assurance should be deleted too.
Response 2:Deleted gap analysis, quality assurance
Point 3 Summary: Line 14: the word tackle should be replaced by improve.
Response 3: Summary line the word tackle has been replaced by improve
Point 4: Page 6 last paragraph is not clear.
Response 4: Please see paragraph 2 of section 2.1
Point 5: Authors are advised to double check the figures of table 5.
Response 5: Checked
Table 6: the presented figures are the average and not the weights.
Response 6: The title of Table is ‘unweighted average scores’